# Asperphenyltones A and B: New Phenylfuropyridinone Skeleton from an Endophytic *Aspergillus* sp. GXNU-A1

**DOI:** 10.3390/molecules27238160

**Published:** 2022-11-23

**Authors:** Jiguo Huang, Xianglong Bo, Furong Wu, Meijing Tan, Youquan Wei, Lixia Wang, Junqiang Zhou, Guiming Wu, Xishan Huang

**Affiliations:** 1School of Chemical Engineering and Technology, Guangdong Industry Polytechnic, Guangdong Engineering Technical Research Center for Green Household Chemicals, Guangzhou 510275, China; 2State Key Laboratory for Chemistry and Molecular Engineering of Medicinal Resources, College of Chemistry and Pharmaceutical Sciences, Guangxi Normal University, Guilin 541000, China

**Keywords:** *Aspergillus* sp., mangrove endophytic fungus, asperphenyltone A, anti-inflammatory effects

## Abstract

Chemical investigation of the fermentation extract of the mangrove endophytic fungus *Aspergillus* sp. GXNU-A1, isolated from *Acanthus ilicifolius* L., discovered an undescribed pair of enantiomers (asperphenyltones A and B (**±1**)), together with four previously described metabolites: nodulisporol (**2**), isosclerone (**3**), 2,3,4-trihydroxy-6-(hydroxymethyl)-5-methylbenzyl alcohol (**4**), and 4,6-dihydroxy-5-methoxy-7-methyl-1,3-dihydroisobenzofuran (**5**). Analyses of the 1D and 2D NMR spectroscopic data of the compounds supported their structural assignments. The presence of the asperphenyltones A and B, which are a pair of enantiomers, was established by HR-ESI-MS, 1D and 2D NMR data and confirmed by single-crystal X-ray diffraction analysis. Metabolites **1**–**5** were evaluated for their anti-inflammatory effects on the production of nitric oxide (NO), and **1**, **3**, and **4** showed significant potential inhibitory activities against NO production in activated macrophages with IC_50_ values of 26–40 μM, respectively.

## 1. Introduction

Marine fungi live in special marine environments, and their secondary metabolic pathways are very different from those of terrestrial fungi. Thus, metabolites of marine fungi are a promising source of active compounds [1,2]. Over the past ten years, mangrove endophytic fungi have been recognized as a potential source of pharmacologically important secondary metabolites [1,2]. Many secondary metabolites with unique structures and prominent biological activities from various marine-derived genera, *Aspergillus* and *Penicillium,* have been reported. [1]. For example, citridone A–C are rare derivatives of phenyl *R*-furopyridone, and have been reported from the cultured broth of *Penicillium* sp. FKI-1938 [3,4]. The phenyl *R*-furopyridone analogs, which represent a class of fungal metabolites with a core structure consisting of an *R*-furopyridone fused with a phenyl moiety, are an important source of structurally new and biologically active alkaloids [3,4,5,6,7]. These phenyl *R*-furopyridone analogs are produced from various fungal sources, such as *Penicillium*, *Aspergillus*, and *Phomopsis* [3,4,5,6,7]. Among them, phenyl *R*-furopyridone analogs from the marine-derived fungi of the genus have attracted considerable attention for their diverse structural complexity and promising bioactivities. For example, a series of phenyl *R*-furopyridone analogs (citridone A–C, from the *Penicillium* sp. FKI-1938) potentiates antifungal miconazole activity against *Candida albicans* [3,4], (±) citridone E shows cytotoxic activities against SF-268, MCF-7, HepG-2 and A549 cell lines with IC_50_ values of 32.0, 29.5, 39.5 and 33.2 µM, respectively [7], and citridone I shows moderate inhibitory activity for nitric oxide production with an IC_50_ value of 52.5 μM [6].

We have reported the isolation of a series of new compounds from the mycelial extract of mangrove endophytic fungi isolated from *Acanthus ilicifolius* L., such as 2-hydroacetoxydehydroaustin, asperlactone A, and guhypoxylonols A–D [8,9,10,11]. In our chemical investigation of this fungus, *Aspergillus* sp., GXNU-A1, a new phenyl *R*-furopyridone derivative **1**, with four known metabolites **2**–**5**, is discovered (Figure 1). The structures of the isolated compounds are established by HR-ESI-MS and 1D and 2D NMR techniques, and this data is compared to the literature data. Herein, the isolation, structure determination, and anti-inflammatory activity of compounds **1**–**5** are described in detail.

## 2. Results and Discussion

Compound **±1** was isolated from the culture liquid, extracted with EtOAc, isolated from the HPLC and obtained as white crystals with the molecular formula C_19_H_19_NO_3_ as determined by HR-ESI-MS, which has a hydrogen adduct quasimolecular ion at *m/z* 310.1442 [M + H]^+^ (calculated 310.1443 for C_19_H_20_NO_3_). The IR absorption bands at 3383, 1694 and 1626 cm^−1^ revealed the presence of amino, carbonyl, and olefinic functionalities. It exhibited UV maximum absorption bands for a pyridone alkaloid at 235 and 323 nm [3,4,5,6,7]. The ^1^H NMR spectrum (Appendix A) showed resonances for five aromatic protons of 1-substituted benzene at *δ*_H_ 7.26–7.51 (5H, m), one olefinic proton of a trisubstituted alkene at *δ*_H_ 6.26 (1H, s, H-10), a methylene group at *δ*_H_ 3.16 (H-4a) and 2.56 (H-4b), and three methyls at *δ*_H_ 2.07 (H-17), 1.66 (H-15) and 1.35 (H-16). The ^13^C NMR and distortionless enhanced polarization transfer spectra showed resonances for three methyl carbons, one methylene carbon, six aromatic or olefinic carbons, and nine quaternary carbons (for NMR data, see Appendix A). The spin system 7.51/7.36/7.26 was successfully established and attributed to the ^1^H–^1^H COSY correlations. The heteronuclear multiple bond correlations (HMBCs) of protons from *δ*_H_ 7.51 (H-12) to C-14 and C-10 were observed, which indicates that **1** had a pent-substituted aromatic ring (fragment A). The HMBCs from *δ*_H_ 10.89 (OH) to *δ*_C_ C-7, C-8, C-9 and *δ*_H_ 9.56 (NH-) to C-7, C-8, C-9, and C-10 suggested that there was a nitrogen five-membered ring, i.e., fragment B. The HMBCs from *δ*_H_ 6.27 (1H, H-10) to *δ*_C_ 129.1 (C-12), 159.7 (C-8) suggested that fragment B was connected to position 1 of fragment A. Meanwhile, the HMBCs from *δ*_H_ 1.66 (H-15) to C-1, and C-3, *δ*_H_ 1.35 (H-16) to C-2, and C-4, from *δ*_H_ 3.16 (H-4a), 2.56 (H-4b) to C-1, C-2, C-3, C-5, and C-15, suggested that it forms another five-membered ring, i.e., fragment C. The HMBCs from *δ*_H_ 2.07 (H-17) to C-1, C-4, and C-7, indicated that fragment C was situated at position 7 of fragment B. The overall analysis of the 1D and 2D NMR data permitted the structural assignment for **1,** as shown in Figure 1. It had highly similar data to citridone C [3,4,5,6,7], except for the presence of a hydroxyl group instead of an olefinic hydrogen at position 10 (Table 1, Figure 2), which results from the dehydration of citridone C during the extraction or purification artifact.

There is only one stereogenic carbon in **1**, and its absolute configuration of C-5 was determined by experimental and calculated ECD (Figure 3). However, the experimental ECD results showed an almost straight line without any absorption peaks, which indicates **1** should be a pair of enantiomers. The structure of **1** was further confirmed via single-crystal X-ray diffraction (Figure 4); its structure was assigned as a pair of enantiomers, and the double bond at C-9 and C-10 was identified as the Z-configuration. Unfortunately, they had not been isolated from each other due to only 1.3 mg of **1** being purified and tested. Therefore, the *R* and *S* configurations of **1** were named asperphenyltone A (**+1**) and asperphenyltone B (**−1**), respectively.

The remaining known compounds **2**–**5** were determined by the analysis of their NMR (Appendix A) data and comparing these with previously published data in the literature. They were identified as nodulisporol (**2**) [12], isosclerone (**3**) [13], 2,3,4-trihydroxy-6-(hydroxymethyl)-5-methylbenzyl alcohol (**4**) [14], and 4,6-dihydroxy-5-methoxy-7-methyl-1,3-dihydroisobenzofuran (**5**) [15].

The phenyl *R*-furopyridone analogs citridones H-L showed inhibitory activity for nitric oxide production (NO) [6]. The compounds (**±1**) have a similar structure to citridones, which may also possess inhibitory activities toward NO. Thus, compounds **±1**–**5** were evaluated for their anti-inflammatory effects on the production of nitric oxide (NO) in the RAW 264.7 macrophage cell line that was exposed to the inflammatory stimulus by lipopolysaccharide (LPS) (Table 2). The results show that compound **1** has a potent inhibitory effect on NO release (IC_50_ 21 μM), while compounds **3**–**4** show weak inhibitory activities for NO production. Compound **2** had no anti-inflammatory properties under its safe concentration, which reveals that **1** may play a crucial role in anti-inflammatory activities.

Biogenetically, pyridone analogues were probably biosynthesized based on the compound (**i**) by cyclization and methyl migration of (**i-1**) (Figure 5). The compound (**i-2**) was oxidated, and constricted the six-membered ring to a five-membered ring, conforming asperphenyltone A (**+1**) and asperphenyltone B (**−1**), further oxidation of asperphenyltone A (**+1**) converted it to citridone C [3,4,5,6,7].

## 3. Experimental

### 3.1. General Experimental Procedures

ECD and UV data were recorded using a JASCO J-715 spectropolarimeter (Jasco, Tokyo, Japan). Single-crystal data were measured on an Oxford Gemini S Ultra diffractometer (Oxford Instrument, Oxfordshire, UK). IR spectra were measured on a Bruker Vector 22 spectrophotometer (Bruker, Billerica, MA, USA) using KBr pellets. One-dimensional and two-dimensional NMR spectra were obtained at 400 MHz for ^1^H and 100 MHz for ^13^C, respectively, on a Bruker Avance III HD 400 spectrometer (Bruker, Ettlingen, Germany) with residual solvent peaks as references. ESI-MS and HR-ESI-MS were obtained on a Bruker Esquire 3000plus and a Waters/Micromass Q-TOF-Ultima (Waters, Milford, MA, USA) mass spectrometers, respectively. Silica gel (300–400 mesh, Qingdao Haiyang Chemical Co. Ltd., Qingdao, China), Sephadex LH-20 (Pharmacia Biotech AB, Uppsala, Sweden), and ODS-A-HG reversed-phase silica gel (12 nm S-50 μm, YMC Co., Ltd., Japan) were used for column chromatography (CC). Silica gel HSGF254 (Yantai Jiangyou Guijiao Kaifa Co., Yantai, China) was used for TLC. Semi-preparative HPLC was performed using an Agilent 1260 HPLC system, and samples were separated on a Waters SunFire-C_18_ column (5 μm, i.d. 10 mm × 250 mm).

### 3.2. Fungal Material and Fermentation

The fungus was isolated from mangrove Acanthus ilicifolius L. leaves collected at the seaside of Qinzhou, Guangxi Province, China, in October 2019. The fungus strain GXNU-A1 was determined as an Aspergillus sp. by 18S rDNA analysis (GenBank accession number: MT626059). The strain was statically cultured at 28 °C for 30 days in 500 mL Erlenmeyer flasks (400 × 200 mL, a total of 80 L), each containing 200 mL of cultural media (glucose 20 g, yeast extract 2 g, MgSO_4_·7H2O 0.1 g, KH2PO4 0.2 g, sea salt 5 g in 1 L water).

### 3.3. Extraction and Isolation

The culture liquid (55 L) was filtered and extracted with EtOAc three times and then concentrated under vacuum to remove the EtOAc to obtain 8.6 g of metabolite extract. The mycelium was extracted by MeOH and then concentrated under vacuum. The concentrate was extracted with EtOAc and then combined with broth extract. The total extract (6.4 g) was fractionated by silica gel CC eluted on a gradient from dichloromethane (D)-methanol (M) (D/M from 100:1 to 70:30) to yield six fractions (Frs.1–6, fraction 1 and 2 were obtained from D/M 10%, fraction 3 from 20%, fraction 3 and 4 from 30%, fraction 5 from 40% and fraction 6 from 50%), based on TLC analysis. Fr.3 was subjected to repeated CC and then purified by semi-preparative HPLC (10 ID × 250 mm, 4.0 mL/min, 60% MeOH in H_2_O) to afford compounds 1 (1.3 mg), 3 (colorless crystal, 2.5 mg) and 4 (colorless crystal, 10.1 mg). Fr.4 was purified further by CC and semi-preparative HPLC (70% MeOH in H_2_O) to obtain compound 2 (colorless crystal, 4.9 mg). Fr.5 was purified further by semi-preparative HPLC (66% MeOH in H_2_O) to obtain compound 5 (colorless crystal, 8.3 mg).

#### Physicochemical and Spectral Data

Compound 1: white crystals; ^1^H and ^13^C NMR data, see Appendix A; (+)-HR-ESI-MS *m/z* 310.1442 [M + H]^+^ (calculated 310.1443 for C_19_H_20_NO_3_).

Crystal data for compound **1**: C_76_H_76_N_4_O_12_ (*Mr* = 222.19 g/mol), monoclinic, space group P2_1_/c (no. 14), *a* = 9.54190(10) Å, *b* = 14.6110 (3) Å, *c* = 11.6889 (2) Å, *β* = 103.789 (2)°, *V* = 1582.68 (5) Å^3^, *Z* = 1, *T* = 103 K, *μ* (Cu K*α*) = 1.54184 mm^−1^, *Dcalc* = 7.503 g/cm^3^, 9440 reflections measured (4.586° ≤ 2Θ ≤ 52.928°), 2112 unique (*R*_int_ = 0.0381, *R*_sigma_ = 0.0979) which were used in all calculations. The final *R*_1_ was 0.0406 (I > 2σ(I)), and *wR*_2_ was 0.0995 (all data). The CCDC Number is 2215742.

### 3.4. Anti-Inflammatory Assay

The anti-inflammatory effects of all compounds were examined for the production of nitric oxide (NO) in LPS-stimulated cells using a method according to our previously described method [16].

## 4. Conclusions

The chemical investigation of the mangrove endophytic fungus *Aspergillus* sp. GXNU-A1 isolated a new pair of phenyl *R*-furopyridone derivatives: asperphenyltone A and B, together with four known metabolites **2**–**5**. Compounds **1**–**5** were evaluated for their anti-inflammatory effects on the production of NO, and compound **1** significantly reduced the production of NO in LPS-stimulated cells with an IC_50_ value of 21 μM.

## Figures and Tables

**Figure 1 molecules-27-08160-f001:**
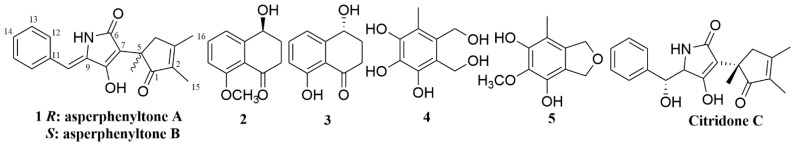
Structures of **1**–**5**.

**Figure 2 molecules-27-08160-f002:**
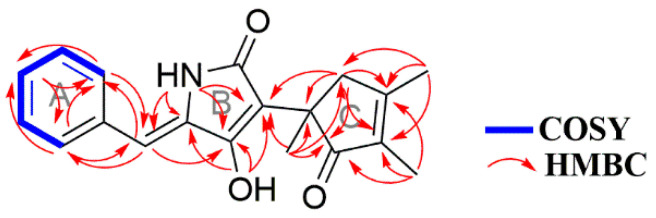
Key correlations of compound **1**.

**Figure 3 molecules-27-08160-f003:**
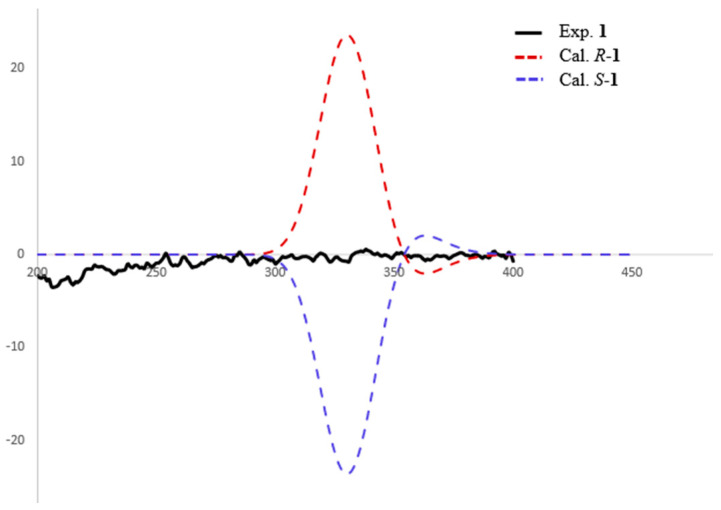
Experimental and calculated ECD spectra of compound **1** (in MeOH).

**Figure 4 molecules-27-08160-f004:**
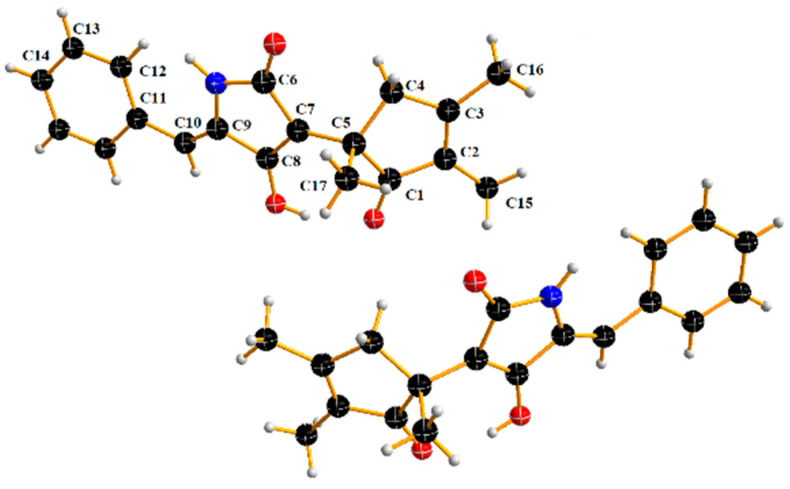
ORTEP diagram showing the structure of a racemate (**±**)-**1** in the crystal.

**Figure 5 molecules-27-08160-f005:**
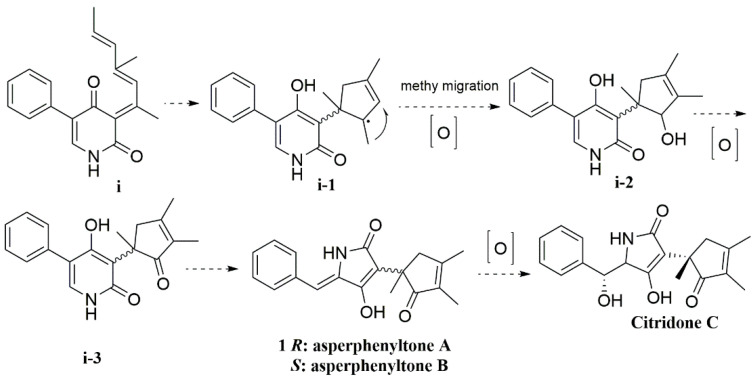
Possible biosynthetic pathways of (**±**)-**1**.

**Table 1 molecules-27-08160-t001:** NMR data of (±) **1** (400 MHz, DMSO-*d*_6_, *δ* in ppm).

Position	*δ*_C_ Type	*δ*_H_, Hz	HMBC
1	213.5, C		
2	131.5, C		
3	171.6, C		
4	46.2, CH_2_	2.56 (1H, d, *J* = 19.0) 3.16 (1H, dd, *J* = 19.0, 3.8)	C-2, C-3, C-5, C-7, C-17
5	46.8, C		
6	171.7, C		
7	105.6, C		
8	159.7, C		
9	134.2		
10	105.5, CH	6.27 (1H, br s)	C-2, C-8, C-16
11	127.4		
12	129.1, CH	7.51 (2H, d, *J* = 7.4)	C-14, C-16
13	128.8, CH	7.36 (2H, t, *J* = 7.4)	C-11, C-15
14	127.5, CH	7.26 (1H, d, *J* = 7.4)	C-12, C-16
15	8.0, CH_3_	1.66 (3H, s)	C-1, C-2, C-3
16	17.1, CH_3_	2.07 (3H, s)	C-2, C-3, C-4
17	25.2, CH_3_	1.35 (3H, s)	C-1, C-4, C-5, C-7
-OH -NH		10.89 (1H, br s) 9.56 (1H, br s)	C-7, C-8, C-9 C-7, C-8, C-9, C-10

**Table 2 molecules-27-08160-t002:** Anti-inflammatory effects of compounds **1**–**5** on the production of NO in lipopolysaccharide (LPS)-stimulated RAW264.7 cells.

Compounds	NO Inhibitory Effects ^a^
**(** **±)−1** **2**	21 >80
**3**	29
**4**	40
**5**	>80
Dexamethasone	38

**^a^** Values present mean ± SD of triplicate experiments.

## Data Availability

Not applicable.

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
