# Peer review of "Asperphenyltones A and B: New Phenylfuropyridinone Skeleton from an Endophytic Aspergillus sp. GXNU-A1"

_molecules, 2022, doi:10.3390/molecules27238160_

Round 1

Reviewer 1 Report

The publication entitled “Asperphenyltones A and B…” by Jiguo Huang and collaborators reports the structure and in-vitro anti-inflammatory action of a new fungal metabolite and of four other known ones. The manuscript is globally well written and presented. Taking into account the following remarks and answering the questions could help to improve the manuscript.

In the title, “derivatives” should be replaced by “skeleton” or “substructure”.

The list of authors ends with “and”, as if an author name were missing.

Four significant figures in IC50 values are two much. Two are sufficient.

Line 14. How do you know that a compound is a new one?

Lines 44 – 66. There is no need to enumerate here what is also reported in Table 1.

Figure 2. What is the difference between a key HMBC correlation and a non-key HMBC correlation? They are all important. Please report them all in Table 1. Did you observe HMBC correlations from OH and NH? Figure S5 shows them and they confirm the proposed structure. As a non-restrictive advice, please consider that free automatic structure elucidation software can help you to check for alternative structures that would fit with 2D NMR data (https://nuzillard.github.io/LSD/). You are also advised to upload your raw NMR data and spectra to a free repository (see doi: 10.1039/C7NP00064B).

Table 1. Chemical shift for C-17 is not reported at the right place. Please append the missing “C” after the chemical shift of C-9 and C-11. How are referenced the 1D spectra? 2.5 ppm for 1H and 39.52 ppm for 13C?

Line 65. Compound 1 is therefore like citridone C but dehydrated. Could it be an extraction or a purification artefact? Probably not but please discuss briefly this possibility.

Line 72. X-Ray data. Figure 4 does not give an indication about the nature of the analysed crystal. Is it a racemate (most likely) or a crystal selected from a racemic conglomerate?

Please report for all compounds the usual structural descriptors such as molecular formula, isomeric SMILES, InChI and InChIKey. The software used for the drawing of the structures can certainly help to do it easily. This can be written in the Supplementary information file.

Line 85. “The 2 had no anti-inflammatory properties”. Is this what you mean?

Compound 5 should be 4,6-dihydroxy-5-methoxy-7-methyl-1,3-dihydroisobenzofuran according to doi: 10.3390/md19070395 (compound 7) but not 4,6-dihydroxy-5-methoxy-7-methylphthalide.

Line 139. Please check X-Ray data and compare with those in the Supplementary information file. One can doubt that Dcalc = 7.503 g/cm3 and that Z=1. Could you report exactly what is present in the unit cell? The SI file says C76  H76 N4 O12 with Z=1, which is strange. Does the unit cell contain two molecules of compound (+)-1 and 2 molecules of compound (-)-1? CCDC has no compound 2215742.

Author Response

Dear Editors:

Herein we resubmit our manuscript entitled as “Asperphenyltones A and B: New Phenylfuropyridinone skeleton from an Endophytic Aspergillus sp. GXNU-A1” (Manuscript ID: molecules-2026804). We have made detailed modification and reply to the comments made by the two review experts. Additional IR and UV data requested by the experts are also discussed in the text and added in the supplementary materials. At the same time, the Extensive English editing of the text was edited and modified, and the certificate was submitted.

The following is a one-by-one responses to Review Expert 1. Thank you!

Comments and Suggestions for Authors

The publication entitled “Asperphenyltones A and B…” by Jiguo Huang and collaborators reports the structure and in-vitro anti-inflammatory action of a new fungal metabolite and of four other known ones. The manuscript is globally well written and presented. Taking into account the following remarks and answering the questions could help to improve the manuscript.

In the title, “derivatives” should be replaced by “skeleton” or “substructure”.

Response: thank you for your suggestion. The “derivatives” was revised as “skeleton” and highlighted.

The list of authors ends with “and”, as if an author name were missing.

Response: thank you for your suggestion. The list of authors were revised and highlighted.

Four significant figures in IC50 values are two much. Two are sufficient.

Response: thank you for your suggestion. All IC50 values were revised as two significant figures and highlighted.

Line 14. How do you know that a compound is a new one?

Response: thank you for your suggestion. This sentence was revised as “…resulted in the isolation of an undescribed pair of enantiomers….”  Usually a search in the scifinder database finds compounds that do not have this structure, So in abstracts we usually describe it as a new compound.

Lines 44 – 66. There is no need to enumerate here what is also reported in Table 1.

Response: thank you for your suggestion. The sentences of Lines 44 – 66 was revised and and highlighted.

Figure 2. What is the difference between a key HMBC correlation and a non-key HMBC correlation? They are all important. Please report them all in Table 1. Did you observe HMBC correlations from OH and NH? Figure S5 shows them and they confirm the proposed structure. As a non-restrictive advice, please consider that free automatic structure elucidation software can help you to check for alternative structures that would fit with 2D NMR data (https://nuzillard.github.io/LSD/). You are also advised to upload your raw NMR data and spectra to a free repository (see doi: 10.1039/C7NP00064B).

Response: thank you for your suggestion. All HMBC correlations were added in Table 1. All observe HMBC correlations from OH and NH were added in Table 1 and figure 2. We carefully check for alternative structures by free automatic structure elucidation software and revised the figures1 and 2. And we uploaded our raw NMR data and spectra to a free repository.

Table 1. Chemical shift for C-17 is not reported at the right place. Please append the missing “C” after the chemical shift of C-9 and C-11. How are referenced the 1D spectra? 2.5 ppm for 1H and 39.52 ppm for 13C?

Response: thank you for your suggestion. It was an error and Chemical shift for C-17 was revised in the table 1. And the 1D spectra were checked carefully and revised.

Line 65. Compound 1 is therefore like citridone C but dehydrated. Could it be an extraction or a purification artefact? Probably not but please discuss briefly this possibility.

Response: thank you for your suggestion. Yes, It does have the possibility of a purification artefact, it only like citridone C but dehydrated, this phenomenon is common in both an extraction and a purification artefact. It was revised and highlighted in the text.

Line 72. X-Ray data. Figure 4 does not give an indication about the nature of the analysed crystal. Is it a racemate (most likely) or a crystal selected from a racemic conglomerate?

Please report for all compounds the usual structural descriptors such as molecular formula, isomeric SMILES, InChI and InChIKey. The software used for the drawing of the structures can certainly help to do it easily. This can be written in the Supplementary information file.

Response: thank you for your suggestion. It is a racemate and the title of Figure 4 was revised and highlighted. Information about crystals were already added in the supplemental materials.

Line 85. “The 2 had no anti-inflammatory properties”. Is this what you mean?

Response: thank you for your suggestion. It was revised and and highlighted.

Compound 5 should be 4,6-dihydroxy-5-methoxy-7-methyl-1,3-dihydroisobenzofuran according to doi: 10.3390/md19070395 (compound 7) but not 4,6-dihydroxy-5-methoxy-7-methylphthalide.

Response: thank you for your suggestion. It was revised as “4,6-dihydroxy-5-methoxy-7-methyl-1,3-dihydroisobenzofuran” and and highlighted.

Line 139. Please check X-Ray data and compare with those in the Supplementary information file. One can doubt that Dcalc = 7.503 g/cm3 and that Z=1. Could you report exactly what is present in the unit cell? The SI file says C76  H76 N4 O12 with Z=1, which is strange. Does the unit cell contain two molecules of compound (+)-1 and 2 molecules of compound (-)-1? CCDC has no compound 2215742.

Response: thank you for your suggestion. The unit cell contain two molecules of compound (+)-1 and of compound (-)-1. It was revised as C76  H76 N4 O12.  We have successfully applied for the preservation number 2215742 from CCDC and checked again. Now we don't know why we can't find its information, and there may be a certain time delay. The CCDC reply said that , “After publication your data will be made available through our joint Access Structures service.” This was the email we accepted following.

Reviewer 2 Report

The current MS can not be published in its present form. The below issues should be carefully addressed.

1- Extensive English editing is needed. There are many typing and grammatical mistakes.

2- The title should be modified, remove enantiomers and add new ``Asperphenyltones A and B: New Phenylfuro-pyridinone derivatives from an Endophytic Aspergillus sp. 3 GXNU-A1``

3- The name of the mangrove should be added to the abstract

5- The names of known metabolites should be added in the abstract

6- Abstract needs revision and rewritten.

7- The introduction is too short and lacks sufficient references.

More about the Aspergillus genus and its bio-metabolites and their bioactivities should be added. Also, the importance of this genus. Authors should focus on the subject of this work.

Also, the phenylfuropyridines should be highlighted in the introduction regarding their chemistry, sources, and biological activity.

8- Lines 34 and 35, authors should mention the reported metabolites and the fungal sources.

9- Are these types of metabolites reported previously from the Aspergillus genus, discuss.

10- In the results and discussion: Brief sentences about the purification of the compounds should be added.

11- The IR and UV spectral data should be added and discussed.

12- The discussion is very weak. For any new compounds, the discussion of all evidence giving by spectral analyses is substantial. All spectral data should be discussed, including COSy, HSQC and their role in the assignment of each substructure, and how the substructures are connected with each other.

13- The structure of citridine C should be added in figure 1.

14- How are the authors detected the enantiomeric nature if these two compounds? Discuss. Is there any observed difference in the spectral data.

15- References that supported the structure assignment should be added.

16- Authors said that they used ECD and X-ray in the structure assignment, however, they didn't discuss them. Also, NOESY has not been discussed.

17- Before the biological part of the discussion, authors should write in brief about inflammation, the role of NO in inflammation, the prevalence of anti-inflammatory agents from marine, and why they selected such activity to test. Are any of the isolated metabolites previously tested as an anti-inflammatory, what are the obtained results, and are these results in alignment with the authors' findings.

Possible biosynthetic pathways of the new metabolites should be included and discussed. What are the authors' suggestions for future studies?

The ECD and Xray analyses instrument are not mentioned in the experimental. Authors should add IR and UV instruments after measuring these parameters.

The plant and fungus genus and species names should be italicized throughout the whole MS.

From which part of the plant the fungus was isolated?. Should be added.

In the extraction, the authors said 7 fractions, and they wrote frs. 1-6!!!!

The polarity of the obtained fractions should be added.

Compound 5 is missing in the extraction.

No need to write the physicochemical characters of the known metabolites, you can add the physical state of them in the extraction.

Author Response

Dear Editors:

Herein we resubmit our manuscript entitled as “Asperphenyltones A and B: New Phenylfuropyridinone skeleton from an Endophytic Aspergillus sp. GXNU-A1” (Manuscript ID: molecules-2026804). We have made detailed modification and reply to the comments made by the two review experts. Additional IR and UV data requested by the experts are also discussed in the text and added in the supplementary materials. At the same time, the Extensive English editing of the text was edited and modified, and the certificate was submitted.

The following is a one-by-one responses to Review Expert 2. Thank you!

Comments and Suggestions for Authors

The current MS can not be published in its present form. The below issues should be carefully addressed.

  • Extensive English editing is needed. There are many typing and grammatical mistakes.
  • Response: thank you for your suggestion. The manuscript was extensive english edited, and the attachment of the certificate was submitted.
  •  
  • The title should be modified, remove enantiomers and add new ``Asperphenyltones A and B: New Phenylfuro-pyridinone derivatives from an Endophytic Aspergillus sp. 3 GXNU-A1``

Response: thank you for your suggestion. The title was revised and highlighted.

  • The name of the mangrove should be added to the abstract

Response: thank you for your suggestion. The name of the mangrove was added to the abstract and highlighted.

  • The names of known metabolites should be added in the abstract

Response: thank you for your suggestion. The names of known metabolites were added to the abstract and highlighted.

  • Abstract needs revision and rewritten.

Response: thank you for your suggestion. Abstract was revised and rewritten.

7- The introduction is too short and lacks sufficient references.

More about the Aspergillus genus and its bio-metabolites and their bioactivities should be added. Also, the importance of this genus. Authors should focus on the subject of this work.

Also, the phenylfuropyridines should be highlighted in the introduction regarding their chemistry, sources, and biological activity.

Response: thank you for your suggestion. The introduction was revised and rewritten.

8- Lines 34 and 35, authors should mention the reported metabolites and the fungal sources.

Response: thank you for your suggestion. metabolites and the fungal sources.were added and reported, and highlighted.

9- Are these types of metabolites reported previously from the Aspergillus genus, discuss.

Response: thank you for your suggestion. Yes, these types of metabolites reported previously from the Aspergillus genus, the introduction was revised, rewritten and highlighted.

10- In the results and discussion: Brief sentences about the purification of the compounds should be added.

Response: thank you for your suggestion. There was the purification of compounds in 3.3. Extraction and isolation, and we added Brief sentences about the purification of the compounds in the results and discussion and highlighted.

11- The IR and UV spectral data should be added and discussed.

Response: thank you for your suggestion. The IR and UV spectral data were added in the SUPPLEMENTARY MATERIAL. And IR and UV spectral data were discussed in the text.

12- The discussion is very weak. For any new compounds, the discussion of all evidence giving by spectral analyses is substantial. All spectral data should be discussed, including COSy, HSQC and their role in the assignment of each substructure, and how the substructures are connected with each other.

Response: thank you for your suggestion. The discussion was rewritten and highlighted.

13- The structure of citridine C should be added in figure 1.

Response: thank you for your suggestion. The structure of citridine C was added in figure 1

14- How are the authors detected the enantiomeric nature if these two compounds? Discuss. Is there any observed difference in the spectral data.

Response: thank you for your suggestion. If there was the enantiomeric nature, we should test this compound experimental ECD, which showed the almost straight line, without any absorption peaks, so we reported the experimental ECD and the calculated ECD, that suggested they were the enantiomeric nature.

15- References that supported the structure assignment should be added.

Response: thank you for your suggestion. References that supported the structure assignment were added 3-7 and highlighted.

16- Authors said that they used ECD and X-ray in the structure assignment, however, they didn't discuss them. Also, NOESY has not been discussed.

Response: thank you for your suggestion. NOESY correlations can only be tested or observed when a compound has more than two chiral carbons, this compound has only a stereogenic carbon, thus it cannot be test or observed. we reported the experimental ECD, which showed the almost straight line, without any absorption peaks, suggested this compound maybe the enantiomeric nature. Then, its structure was further confirmed via single-crystal X-ray diffraction, and they were. This discuss was added in the text and highlighted.

17- Before the biological part of the discussion, authors should write in brief about inflammation, the role of NO in inflammation, the prevalence of anti-inflammatory agents from marine, and why they selected such activity to test. Are any of the isolated metabolites previously tested as an anti-inflammatory, what are the obtained results, and are these results in alignment with the authors' findings.

Response: thank you for your suggestion. Introduction and before the biological part of the discussion was revised and rewritten.

Possible biosynthetic pathways of the new metabolites should be included and discussed. What are the authors' suggestions for future studies?

Response: thank you for your suggestion. Possible biosynthetic pathways of 1 was discussed in the text.  We are now also working to explore the diversity of these derivatives.

The ECD and Xray analyses instrument are not mentioned in the experimental. Authors should add IR and UV instruments after measuring these parameters.

Response: thank you for your suggestion. The ECD, Xray, IR and UV analyses instrument was added in the experimental,

The plant and fungus genus and species names should be italicized throughout the whole MS.

Response: thank you for your suggestion. These errors were revised and highlighted in the text.

From which part of the plant the fungus was isolated?. Should be added.

Response: thank you for your suggestion. This error was revised and highlighted in the text.

In the extraction, the authors said 7 fractions, and they wrote frs. 1-6!!!!

Response: thank you for your suggestion. This error was revised and highlighted in the text.

The polarity of the obtained fractions should be added.

Response: thank you for your suggestion. This error was revised and highlighted in the text.

Compound 5 is missing in the extraction.

 Response: thank you for your suggestion. This error was revised and highlighted in the text.

No need to write the physicochemical characters of the known metabolites, you can add the physical state of them in the extraction.

Response: thank you for your suggestion. This error was revised and highlighted in the text. known metabolites were added the physical state of them in the extraction.

Round 2

Reviewer 2 Report

No comment